# Voluntary wheel running exercise rescues behaviorally-evoked acetylcholine efflux in the medial prefrontal cortex and epigenetic changes in ChAT genes following adolescent intermittent ethanol exposure

**Matthew J. Fecik[1], Polliana T. Nunes[1], Ryan P. Vetreno[2], Lisa M. Savage[1]***

**1** Department of Psychology, Behavioral Neuroscience Area, Binghamton University-State University of New York, Binghamton, NY, United States of America, **2** Bowles Center for Alcohol Studies, School of Medicine, University of North Carolina at Chapel Hill, Chapel Hill, NC, United States of America

* lsavage@binghamton.edu

**Data Availability Statement:** The data are available in the supporting information files.

## Abstract

Adolescent intermittent ethanol (AIE) exposure, which models heavy binge ethanol intake in adolescence, leads to a variety of deficits that persist into adulthood—including suppression of the cholinergic neuron phenotype within the basal forebrain. This is accompanied by a reduction in acetylcholine (ACh) tone in the medial prefrontal cortex (mPFC). Voluntary wheel running exercise (VEx) has been shown to rescue AIE-induced suppression of the cholinergic phenotype. Therefore, the goal of the current study is to determine if VEx will also rescue ACh efflux in the mPFC during spontaneous alternation, attention set shifting performance, and epigenetic silencing of the cholinergic phenotype following AIE. Male and female rats were subjected to 16 intragastric gavages of 20% ethanol or tap water on a two-day on/two-day off schedule from postnatal day (PD) 25–54, before being assigned to either VEx or stationary control groups. In Experiment 1, rats were tested on a four-arm spontaneous alternation maze with concurrent *in vivo* microdialysis for ACh in the mPFC. An operant attention set-shifting task was used to measure changes in cognitive and behavioral flexibility. In Experiment 2, a ChIP analysis of choline acetyltransferase (ChAT) genes was performed on basal forebrain tissue. It was found that VEx increased ACh efflux in the mPFC in both AIE and control male and female rats, as well as rescued the AIE-induced epigenetic methylation changes selectively at the *Chat* promoter CpG island across sexes. Overall, these data support the restorative effects of exercise on damage to the cholinergic projections to the mPFC and demonstrate the plasticity of cholinergic system for recovery after alcohol induced brain damage.

**Funding:** This work was funded by U01AA028710 The funders had no role in study design, data collection and analysis, decision to publish, or preparation of the manuscript.

**Competing interests:** The authors have declared that no competing interests exist.

## Introduction

Adolescence is a period of brain development in which crucial changes occur to various cortical and limbic structures [1–3]. Exposure to binge levels of ethanol intoxication has the potential to induce physiological and behavioral changes that persist into adulthood [4–6]. Heavy binge-level alcohol consumption during adolescence produces persistent effects on both neurocognitive functions as well as brain structure such as alterations to grey volume and white matter integrity—particularly in the frontal cortex [7–10]. Animal models of adolescent alcohol exposure, such as the adolescent intermittent ethanol (AIE) model, replicate heavy binge alcohol intoxication that is reported in some adolescents [11]. Adult rats exposed to the AIE model have been shown to display several deficits in various domains of executive functioning, such as cognitive flexibility, behavioral flexibility, working memory, and impulsivity [12–14]. Some of the deficits in behavior that have been observed following AIE are thought to be due to damage to the basal forebrain cholinergic system [15, 16]. Specifically, it has been shown that AIE globally reduces the expression of choline acetyltransferase positive (ChAT+) in neurons of the basal forebrain [14, 17–19].

The consequences of AIE on the basal forebrain cholinergic system are of particular interest due to these regions' contribution to attention, memory, and cognition [20–22]. There are two main basal forebrain cholinergic circuits that have been investigated following AIE. One consists of the cell bodies in the medial septal nucleus and the ventral arm of the diagonal band (MS/DB) that send cholinergic terminals to both the hippocampus and the cortex, while the other consists of the nucleus basalis of Meynert (NbM) which projects to the entire neocortex [23–25]. In rats, AIE has been shown to reduce behaviorally-evoked ACh efflux in the hippocampus and frontal cortex [19].

Given that the NbM complex, consisting of the NbM, substantia innominata, and horizontal band, send cholinergic projections to the entirety of the cortical mantle, it is not surprising that there are cortical-dependent behavioral deficits following ethanol exposure in adolescent rats. One particular region of interest involved in executive function and top-down control of behavior is the mPFC [26]. This region is important for the regulation of cognitive flexibility as measured by attention set-shifting [27]. AIE-treated rats have been shown to have impaired cognitive flexibility as measured by an operant set-shifting task in which the animal must shift from using a spatial cue to a visual one [18, 28]. AIE in early, but not late adolescence, has been shown to decrease cognitive flexibility as measured by the number of regressive errors, but not perseverative errors, when shifting extra dimensionally from a visual cue to a spatial cue [29], suggesting that cognitive flexibility deficits may differ as a function of task demand and age at alcohol exposure. In addition, AIE has been shown to cause impairments to reversal learning [30, 31], which is dependent on the orbital frontal cortex (OFC) another region that receives cholinergic signals from the NbM complex [32, 33]. Taken together with data on cognitive and behavioral flexibility, this suggests that AIE broadly affects the cholinergic circuits originating in the NbM and projecting to the cortical mantle.

Data suggests that the mechanism underlying alcohol-related damage to the cholinergic system likely involves a reduction in the content of neurotrophins such as NGF and BDNF as well as their receptors TrkA and TrkB, respectively [34–36]. This reduction in neurotrophic activity may be driving an epigenetic silencing of the cholinergic phenotype, leading to a reduction in cholinergic transmission. Additionally, there is data suggesting that induction of neuroimmune signaling drives the loss of hippocampal neurogenesis [37] and ChAT+ cholinergic neurons in the basal forebrain, specifically HMGB1 neuroimmune signaling and REST-G9a gene repression [38]. Fortunately, there is strong evidence to suggest that this type of pathology can be recovered in a time-dependent manner [39, 40]. Voluntary wheel running

exercise has been shown to prevent AIE-induced suppression of the cholinergic neuron phenotypic in the basal forebrain when used either during or after AIE treatment [31, 41]. In addition, voluntary wheel running has been shown to decrease methylation of ChAT promoter regions, signifying its ability to regulate the expression of cholinergic phenotype via epigenetic mechanisms [31].

However, the question of whether such plasticity exists in the NbM-PFC circuit, in both male and females, following AIE has not yet been answered. In the current study, we explored the hypothesis that voluntary wheel running exercise, independent of sex, will be sufficient to rescue ACh efflux in the mPFC, as well as set shifting behavior and epigenetic suppression of the cholinergic phenotype following AIE.

## Materials and methods

### Experiment 1

**Subjects.** Both male (n = 47) and female (n = 60) Long-Evans rats were bred in house at Binghamton University. A total of 17 litters were used in this experiment and in most cases only 1 pup per sex per litter was assigned to an experimental condition (Sex, Exposure, Exercise); however, in some cases 2 pups of each sex per litter were assigned to an experimental condition. In the interest of controlling for potential litter effects, litter was included as a factor in analyses of all behavioral and microdialysis data. There were no litter effects nor any interactions involving litter on any of these measures (all p's > 0.05), and therefore data are represented here without litter as a factor. Rats were either pair housed or housed with two other same sex conspecifics in a humidity and temperature-controlled colony room set to 20°C. All experimental procedures were approved by the Institutional Animal Care and Use Committee (IACUC) at the State University of New York at Binghamton and followed the National Institutes of Health (NIH) Guide for Care and Use of Laboratory Animals.

**Adolescent intermittent ethanol exposure.** Rats were weaned on postnatal day (PD) 21 in standard housing conditions. Between PD 25–27, depending on the timing of birth dates, rats were randomly assigned to the adolescent intermittent ethanol (AIE) group or a control group and were administered 16 intragastric gavages of either 20% EtOH (v/v) delivered at a dose of 5.0 g/kg, or tap water, on a 2-day on/off cycle in which rats were dosed once per day for 2 days, followed by a 2-day recovery period until PD 55–57 [17, 31, 41, 42]. Blood samples were collected an hour following the eighth gavage and an AM1 Alcohol Analyzer (Analox Instruments, Stourbridge, UK) was used to analyze samples and determine blood ethanol content (BEC). Fig 1A displays the experimental procedures and timeline for Experiment 1.

**Cannulation implantation surgery.** Approximately 1–2 weeks after the cessation of AIE treatment, cannulation surgeries were performed. A ketamine (85 mg/kg)/dexmedetomidine (0.04 mg/kg) mixture at a dosage of 1.0 mL/kg was administered intraperitoneally to induce anesthesia. Subjects were placed in a stereotaxic apparatus (David Kopf Instruments, Tujunga, CA) and 5 mm guide cannulae (Synaptech Technology Inc., Marquette, MI) were placed at the coordinates AP = + 2.7, ML = +/− 0.7, and DV = − 3.0 mm relative to bregma, corresponding to the medial prefrontal cortex [43]. Cannula placement was counterbalanced such that some animals had cannulations in the left hemisphere and some in the right hemisphere. The guide cannulae were fixed to the head using acrylic dental cement anchored by bone screws inserted into the skull. Rats were awoken using a 0.1 mL dose of atipamezole (Amerisource Bergen MWI Animal Health, Boise, ID) administered intraperitoneally. Carprofen (5.0 g/kg; Zoetis, Kalamazoo, MI) was administered immediately following surgery, as well as 24-hours and 48-hours post-surgery as an analgesic. Rats were closely monitored and allowed to recover for at least 7 days following surgery, prior to exposure to the exercise paradigm.

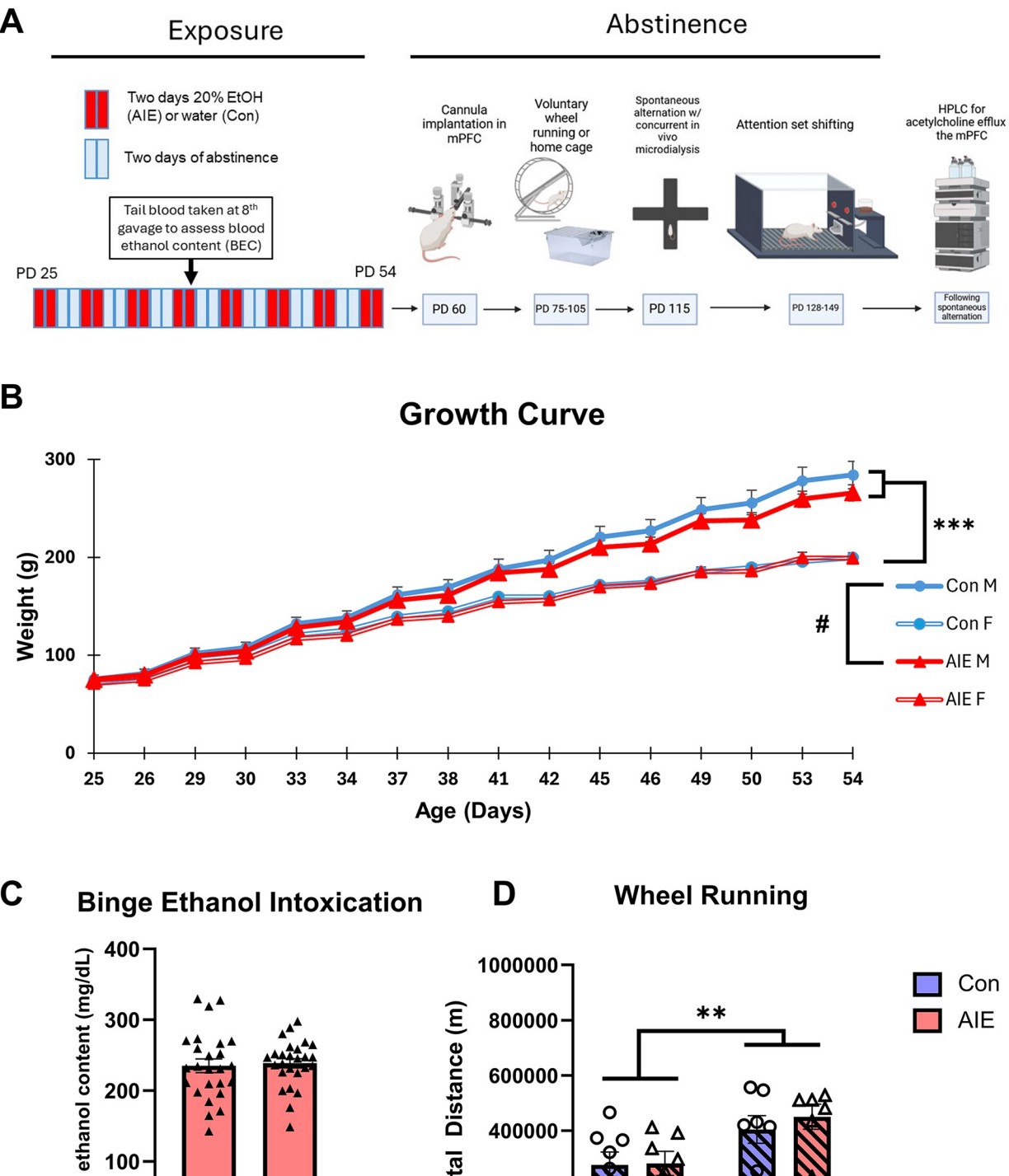

**Fig 1. Experiment 1 treatment timeline, adolescent intermittent ethanol treatment, and voluntary wheel running exercise.** Subjects received intragastric gavages of either EtOH (5g/kg) or tap water on a two-day-on two-day-off schedule from PD 25–54. Blood ethanol content in AIE animals was measured 1h following the eighth gavage. At PD ~60, subjects were implanted with a guide cannula into the mPFC and were subsequently assigned to either cages with free access to a running wheel (VEx) or a standard cage (Stat) from PD 75–105. At PD ~115, subjects were run on a spontaneous alternation task with concurrent *in vivo* microdialysis to measure behaviorally evoked ACh efflux. Following the conclusion of

microdialysis for all subjects, they were run on an operant attention set shifting task to measure cognitive and behavioral flexibility (A). Growth Curve for AIE (Male = 25, Female = 26) and control (Male = 22, Female = 34) rats during gavage. Weights increased over time, while males weighed more than females across all time points and gained weight at faster rate (B). Blood ethanol content for male (n = 25) and female (n = 26) AIE subjects. All subjects reached the criteria for binge levels of intoxication (80mg/dL) and BEC did not differ as a function of sex (C). Total wheel running distances among VEx animals (each data point is one cage) (D). There was a main effect of Sex: Females ran significantly more than males.

**Exercise paradigm.** AIE- and water-treated control rats were assigned randomly into one of two conditions: a voluntary exercise condition or a stationary control condition. Rats in the VEx condition were pair housed in a cage with an exercise wheel (35.56 cm diameter; turning resistance < 6 gms) attached to a clear polycarbonate home cage (Lafayette Instruments, Lafayette, IN, 48.3 × 26.7 × 20.3 cm). Rats in the stationary control condition were pair or triple housed and left in their home cage. In the VEx condition, the wheels had an infrared sensor to measure daily wheel revolutions via Scurry software (Lafayette Instruments) running on a laptop adjacent to the running wheel setup. Rats in the VEx condition were allowed to run on the wheels unrestricted starting at PD 75 for 30 days before being moved back to standard home cages at PD 105.

**Behavioral testing.** *Spontaneous alternation with concurrent in vivo microdialysis*. In Experiment 1, subjects underwent behavioral testing approximately one week following respective stationary/VEx exposure (see Fig 3A). Only animals with detectable ACh levels and accurate cannula placement were included in spontaneous alternation and microdialysis data analysis (Males: Con-Stat = 11, Con-VEx = 7, AIE-Stat = 8, AIE-VEx = 9; Females: Con-Stat = 9, Con-VEx = 9, AIE-Stat = 11, AIE-VEx = 8). On the day of testing, a microdialysis probe (S-8020; 5 mm; Synaptech Inc.) was inserted into the guide cannula, and the rat was acclimated in an opaque white chamber (30 cm × 40 cm × 35 cm) for 60-minutes before testing began. A CMA microinfusion pump (CMA/400 pump; Holliston, MA) connected to the probe and artificial cerebrospinal fluid (7.4 pH solution: 127.6 mM NaCl, 0.9 mM NaH2PO4, 2.0 mM Na2HPO4, 4.0 mM KCl, 1.3 mM CaCl2 dihydrate, 1.0 mM glucose, and 0.9 mM MgCl2) while 250 nM neostigmine hydrobromide (Sigma-Aldrich Corp.) solution was continuously perfused at 2.0 μL/min during the habituation period.

Following the 60-minute habituation period, baseline dialysate samples were collected at three 6-minute intervals to determine the baseline level of ACh efflux into the mPFC. Spontaneous alternation was chosen as an assay to access activity-dependent ACh efflux as both hippocampal and frontal cortical ACh levels are significantly elevated during exploration of the maze [18, 19, 42, 44, 45]. During spontaneous alternation testing, the rat was placed into the center of the plus maze with four arms of 55 cm equidistant lengths with 12 cm high walls. The maze was elevated 80 cm from the floor with numerous extra-maze cues on the walls and around the room. All activity regarding sequence and number of arms entered was recorded while the rat was allowed to freely transverse the maze for a period of 18 minutes during which microdialysis collection continued. An arm entry was defined as having all four paws simultaneously in the arm of the maze, and an alternation is defined as entry into four different arms without the same arm being entered twice during the sequence. For example, if the rat performs the following sequence of arm entries: A, C, D, B, C, A, D, B; the sequence of arm entries 1–4 of ACDB was an alternation, but the sequence of arm entries 2–5, CDBC, was not. The percent alternation score was calculated as $Score = \frac{(number\ of\ alternations)}{(the\ total\ possible\ amount\ of\ alternations)} \times 100$. Animals that made less than 10 arm entries were excluded from analysis. The rat was placed back in the opaque habituation chamber following testing for 18 minutes of post-baseline dialysate collection. Dialysate was then frozen at -20˚C until they were later thawed for HPLC analysis.

*Attention set shifting and reversal learning.* Operant conditioning procedures were emulated from previous work using this task [46]. Initial food restriction to 85% of free feeding weight occurred approximately a week before testing began. Two days prior to testing, all rats were habituated to the sucrose pellets that served as reinforcement (Rodent Purified Dustless Precision Pellet; Bio-Serve, Flemington, NJ). Testing occurred approximately 20 days following exercise, and two weeks following spontaneous alternation. Testing was done using 12 operant chambers (30 cm × 33 cm × 23 cm; Med Associates Inc., St. Albans, VT) housed within sound-attenuating boxes (59 cm × 55 cm × 36 cm), with each box being outfitted with a fan to provide white noise and minimize the effect of ambient sounds. Each chamber had two retractable levers, one on the left side and on the right side of a magazine where the animal collects the reward pellet. A house light was positioned on the opposite wall from the magazine in the top right corner, while two stimulus lights were positioned above each of the levers. Each of the boxes was interfaced to a computer program that records the data of each testing session (MED-PC IV, Med Associates Inc.).

During initial training, each rat learned to press an extended lever on a fixed ratio 1 (FR1) schedule for a food pellet reward with the side of a lever being predetermined randomly for each before testing starts. Both house lights were illuminated during this task, and each session lasted 30 minutes until a stability criterion of 50 lever presses or more was met. The following day, rats were shaped on the opposite lever side to the same criterion. The next day, rats were trained on retractable lever training in which a single lever of a randomly determined side was extended each trial and the rat must press the lever within 10 seconds before it retracted. Criterion was defined as having less than 5 omissions over two consecutive 30-minute sessions. The side of the lever that was presented was pseudorandomized. Lastly, side bias was determined with the final training task in which the rat was presented with both levers over seven trials, and the one that is responded to most was determined to be the preferred side.

Following the determination of side bias, rats were tested on the response side (acquisition task), in which the rat learned to respond to the lever opposite of its original side preference. The house light was illuminated during a trial, and either the left or right stimulus light was randomly illuminated above the levers, which was presented for 10 seconds. Each rat was tested on a minimum of 20 trials, with a maximum of 220 trials if they did not reach the criterion. Criterion consisted of making 10 correct lever presses consecutively. There was an inter-trial interval of 20 seconds. Animals were tested again the following day if they did not reach the criterion. The following day after reaching the criterion, a set shift was introduced (shift to cue task), in which the first 20 trials had the same rule as the previous task, after which the rule switched, and the rat had to learn to respond to the lever that is underneath the illuminated stimulus light with the same criterion as described on the previous task. The following day, the rat was tested on a second set shift task (shift to response), in which the first 20 trials were the same rule as the previous day but afterwards, the cue light needed to be ignored and the lever opposite their original side preference was the reinforcer. The day following the day they reached criterion on the second set shift, a reversal task was introduced in which the first 20 trials are the same rule as the previous day but following which the rat had to switch its response to the lever opposite of what was rewarded in the previous task.

During sessions with a rule change, errors made following the rule shift were classified as either perseverative or regressive based on [46]. In short, the trials during the shift to cue and shift to response phases of the task were divided into bins of 16. If the number of errors made during a bin were less than 5, then all the errors in that bin were labeled perseverative. If 5 or more errors were made in that bin, it was labeled regressive. Once a regressive error was made, all subsequent errors in that session were labeled regressive. For reversal the same criteria applied, except less than 10 errors needed to be made in each bin for it to be labeled regressive.

The number of trials it takes to reach criterion, total errors, perseverative errors, regressive errors, and latency to lever press and take the reward were analyzed for group differences.

**High performance liquid chromatography.** To assess the ACh concentration of the microdialysis samples in the mPFC, high-performance liquid chromatography (HPLC) with electrochemical detection (Amuza, San Diego, CA, USA) was used. The software program Envision (provided by Amuza, San Diego, CA, USA) was used to analyze chromatographs obtained. ACh peaks were quantified by comparison to standard peak heights of 100nm, 20nm, and 4nm solutions. Only measurable ACh levels, 5 femtomoles or greater, were included in the analyses.

**Cresyl violet staining.** One to three weeks following the conclusion of attention set shifting, all experiment 1 rats were deeply anesthetized (Fatal-Plus, Vortech Pharmaceuticals, Dearborn, MI) before being transcardially perfused with ice cold phosphate-buffered saline followed by 4% paraformaldehyde. Brains were extracted and stored in PFA for 24 hours before transfer to 30% sucrose at 4˚C until slicing. Brains were sliced using a sliding microtome at 40 μm thickness and stored in cryoprotectant at -20˚C until cresyl violet staining (FD Neurotechnologies, Columbia, MA, USA). Each microdialysis subject had their cannula placement verified. Fig 3B demonstrates the typical cannula placement in the mPFC. In particular, the prelimbic cortex was targeted due to its cholinergic innervation. Subjects with misplaced cannulae were not included in the final analysis of microdialysis during spontaneous alternation data.

## Experiment 2

**Subjects.** Male and female Sprague Dawley rats were bred in house at Binghamton University. A total of 10 litters for Experiment 2 were produced with most commonly 1 pup per sex of each litter assigned to an experimental condition, but in some instances 2 pups per sex/litter were assigned to an experimental condition. Rats were either pair housed or housed with two other same sex conspecifics in a humidity and temperature-controlled colony room set to 20˚C. All experimental procedures were approved by the Institutional Animal Care and Use Committee (IACUC) at the State University of New York at Binghamton and followed the National Institutes of Health (NIH) Guide for Care and Use of Laboratory Animals.

Fig 2 displays the experimental procedures and timeline for Experiment 2.

**Adolescent intermittent ethanol exposure.** The same ethanol exposure procedures used in Experiment 1 were followed in Experiment 2.

**Exercise paradigm.** The same exercise procedures used in Experiment 1 were followed in Experiment 2.

**Chromatin immunoprecipitation (ChIP).** In Experiment 2, on PD 120, CON- and AIE-treated rats were briefly restrained and sacrificed via rapid decapitation (Decapicones, ThermoFisher Scientific, Waltham, MA, USA). Basal forebrain tissue was dissected according to regions defined in the atlas of Paxinos and Watson [43], rapidly frozen flash frozen in 2-methylbutane (Sigma Aldrich, St. Louis MO) and stored at -80˚C for later ChIP assessment. ChIP was performed as previously described [31, 38, 47, 48]. Briefly, basal forebrain tissue samples were homogenized, fixed in 1.0% methanol-free formaldehyde, quenched with 1.0 M glycine, lysed with lysis buffer (1.0% [v/v] SDS, 10 mM EDTA, 50 mM Tris-HCl [pH 8.0]), and chromatin sheared to fragments of <1000 bp on a Covaris ME220. Input DNA fractions were removed from the sheared chromatin to be processed separately and the remaining sheared chromatin was incubated overnight at 4˚C with an antibody against H3K9me2 (Abcam, Cat. #ab1220). Protein A Dynabeads (ThermoFisher Scientific, Austin, TX) were added and rotated at 4˚C for 1 hr followed by five washes in ChIP wash buffer. Both immunoprecipitated DNA

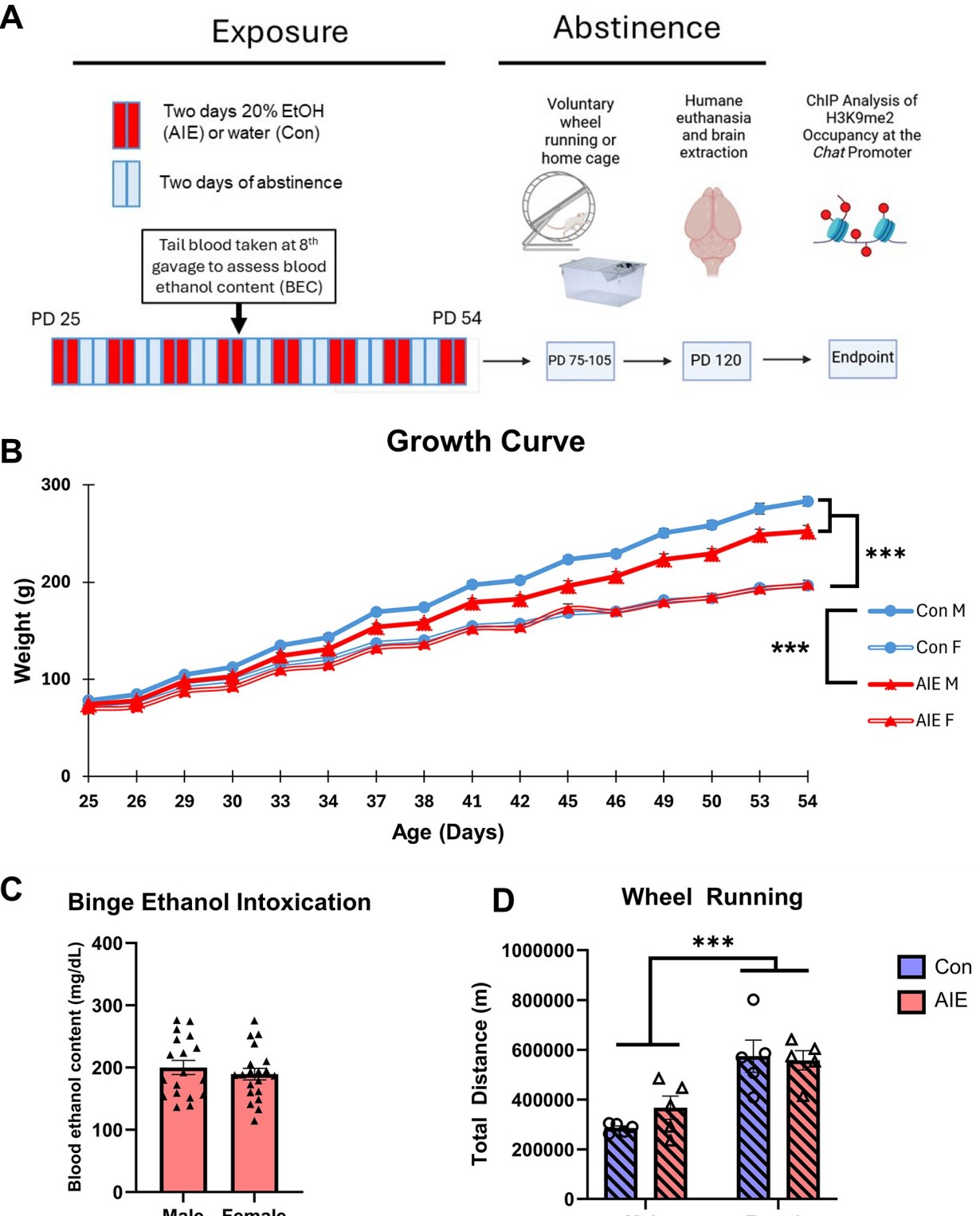

**Fig 2. Experiment 2 treatment timeline, adolescent intermittent ethanol treatment, and wheel running.** Subjects received intragastric gavages of either EtOH (5g/kg) or tap water on a two-day-on two-day-off schedule from PD 25–54. Blood ethanol content in AIE animals was measured 1h following the eighth gavage. After AIE, subjects were assigned to either cages with free access to a running wheel (VEx) or a standard cage (Stat) from PD 75–105. At PD ~120 subjects were humanely euthanized, and brains were flash frozen until basal forebrain punches were taken. Growth Curve for AIE (Male = 19, Female = 20) and control (Male = 20, Female = 20) rats during gavage (**A**). Weights increased over time, while males weighed

more than females across all time points and gained weight at faster rate. Additionally, control males gained weight at a faster rate than AIE males. Blood ethanol content for male (n = 19) and female (n = 20) AIE subjects (**B**). All subjects reached the criteria for binge levels of intoxication (80mg/dL) and BEC did not differ as a function of sex (**C**). Total wheel running distances among VEx animals. (**D**) There was a main effect of Sex: Females ran significantly more than males. *** = p<0.001.

and input DNA were eluted in 10% (w/v) Chelex by boiling at 95˚C for 10 min followed by centrifugation. The resulting DNA was quantified using qPCR with SSOAdvanced Universal SYBR Green Supermix (Bio-Rad, Berkeley, CA) using primers targeted against the *Chat* promoter (Forward `ACTTGATTGCTGCCTCTCTC`; Reverse `GGGATGGTGGAAGATACAGAAG`) and the *Chat* promoter CpG island (Forward `TGCATCTGGAGCTCAAATCGT`; Reverse `GGGGATAGTGGTGACGTTGT`). The ΔΔCt method was used to determine fold change relative to control and was normalized to the Input DNA fraction.

**Statistical analyses.** Analyses were performed in Prism (GraphPad Software, San Diego CA) or SPSS (IBM, Armonk, NY). An unpaired t-test was used to analyze the blood ethanol concentration of male and female AIE animals. A three-way repeated-measures ANOVA was used to analyze animal weights during AIE (Within-subjects: Time; Between-subjects: AIE vs. Control (Exposure); VEx vs. Stat; Exercise). A two-way ANOVA was used to analyze total running wheel distance across 30 days (AIE vs. Control; Male vs. Female). Four-way repeated-measures ANOVAs were used to analyze all attention set shifting measures (Within-subjects: Task; Between-subjects: Exposure; Sex, Exercise). A four-way repeated-measures ANOVA was used to analyze HPLC data (Within-subjects: Sample; Between-subjects: Exposure; Sex, Exercise). ChIP data were analyzed using 2 (Exposure) × 2 (Sex) × 2 (Exercise) ANOVAs. The standard error from the mean is represented by error bars on figures. Outliers for each treatment group across each measure were determined via Grubb's tests (α = 0.05). Outliers were removed where indicated. Mauchley's test for sphericity was used on all ANOVA data, which revealed that data for weight gain, wheel running, microdialysis, and attention set shifting had violated the sphericity assumption (p's < 0.05). Therefore, Greenhouse-Geisser corrections were applied to these data to avoid type I errors.

## Results

### Weight gain

In Experiment 1, there was a trending Exposure × Sex × Time interaction (F[1.652, 171.794] = 6.308, p = 0.061) on weight gain during gavage treatment: Male control subjects gained more weight than male AIE subjects as they aged, but female AIE and female Con subjects gained weight at similar rates (see Fig 1B). Post hoc analyses did not reveal any significant differences between AIE and Con males at any of the 16 exposures (all p's > 0.09). At the time of behavior testing, there were no differences in weight between Con and AIE males (Con = 471.0, AIE = 461.6; p > 0.42). In Experiment 2, there was also a significant Exposure × Sex × Time interaction (F[13.288, 208.110] = 13.288, p < 0.001), such that Con males gained weight quicker than AIE males, but Con and AIE females gained weight at similar rates (see Fig 2B).

### Binge-like blood ethanol concentrations and wheel running

In both experiments, all AIE rats had BECs of a minimum of 114.1 mg/dL and reached well above the criterion for intoxication of 80 mg/dL of EtOH [49] following the eighth gavage (*M* = 218.9). There was no effect of Sex (both p's > 0.75) on BEC in either Experiment 1 (Fig 1C) or Experiment 2 (see Fig 2C). One male rat in experiment 1 had its data excluded from this analysis as there was contamination in its sample.

Analysis of total wheel running distance revealed a main effect of Sex in both Experiment 1 (F[1, 24] = 9.964, p < 0.01; see Fig 1D) and Experiment 2 (F[1, 16] = 29.01, p < 0.001; see Fig 2D), such that cages with females ran significantly more meters than males. There was no effect of Exposure (p's > 0.48) on running distance nor was there a Sex × Exposure interaction (p's > 0.28) in either experiment.

### Experiment 1: Microdialysis and behavior

**Acetylcholine microdialysis.** Analysis of ACh efflux during spontaneous alternation (see Fig 3A) revealed a significant Exposure × Phase interaction (F[5.086, 310.263] = 3.790, p < 0.01), as AIE rats had blunted ACh efflux during maze exploration compared to Con rats (see Fig 3C). Additionally, there was a significant Exercise × Phase interaction (F[5.086, 310.263] = 2.579, p < 0.05), such that VEx animals displayed a larger magnitude of behaviorally-evoked ACh efflux during maze exploration than stationary controls. Additionally, there were significant main effects of Exposure (F[1, 61] = 7.096, p = 0.01) and Exercise (F[1, 61] = 6.888, p < 0.05), such AIE rats had lower ACh efflux and VEx rats had higher ACh efflux overall. There was no main effect of Sex, nor were there any other significant interactions. Three animals had their HPLC data excluded as they were outliers (Final numbers in Fig 3).

**Spontaneous alternation.** Analysis of percent alternation scores did not yield main effects of Exposure, Exercise, or Sex), nor were there any interactions between these variables (all p's > 0.08; see Fig 3D). Analysis of the number of arm entries made during spontaneous alternation (see Fig 3E) revealed a main effect of Sex (F[1, 58] = 11.16, p < 0.001), as females across conditions made more arm entries than males. There was also a main effect of Exercise (F[1, 58] = 6.745, p < 0.05), as animals that exercised made fewer arm entries than animals that did not exercise. However, the number of arm entries was not affected by Exposure, nor was there an interaction between any of these variables (all p's > 0.32). Two animals had their arm entry data excluded as they were outliers (Final numbers in Fig 3).

**Operant attention set-shifting.** Analysis of the number of trials to reach criterion and number of errors made during operant attention set shifting (see Fig 4B and 4C) revealed a main effect of task Phase (Trials: F[1.205, 116.922] = 72.250; Errors: F[1.237, 117.533] = 12.625, both p's < 0.001), as all rats required more trials to reach criterion and made more errors during the shift to cue phase than any other stage of the task. However, there was no effect of Exposure, Sex, or Exercise on the number of trials to reach criterion, nor were there any interactions (all p's > 0.11). Five subjects had their error data excluded and three subjects had their trial data excluded as they were outliers on one or more phases of the task (Final numbers in Fig 4).

Analysis of the number of perseverative errors and regressive errors (see Fig 4D and 4E) made revealed a main effect of Task (Perseverative: F[1.237, 117.533; Regressive: F[1.386, 130.313] = 27.629, p < 0.001] = 12.625, p < 0.001), as all rats made more errors during the shift to cue phase than any other stage of the task. Additionally, there was a significant Exposure X Exercise X Task interaction (F[1.237, 117.533] = 4.363, p < 0.04) such that AIE-Stat and Con-VEx rats made more perseverative errors during the shift to cue, but less perseverative errors during the shift to response compared to AIE-VEx and Con-Stat subjects. Post hoc analyses revealed that AIE-Stat subjects made more perseverative errors than AIE-VEx subjects during the shift to cue phase (p < 0.05), but less during the shift to response phase (p < 0.05). There were no main effects of Exposure, Exercise, or Sex. Five subjects had their perseverative error data excluded and eight subjects had their regressive error data removed as they were outliers on one or more phases of the task (Final numbers in Fig 4).

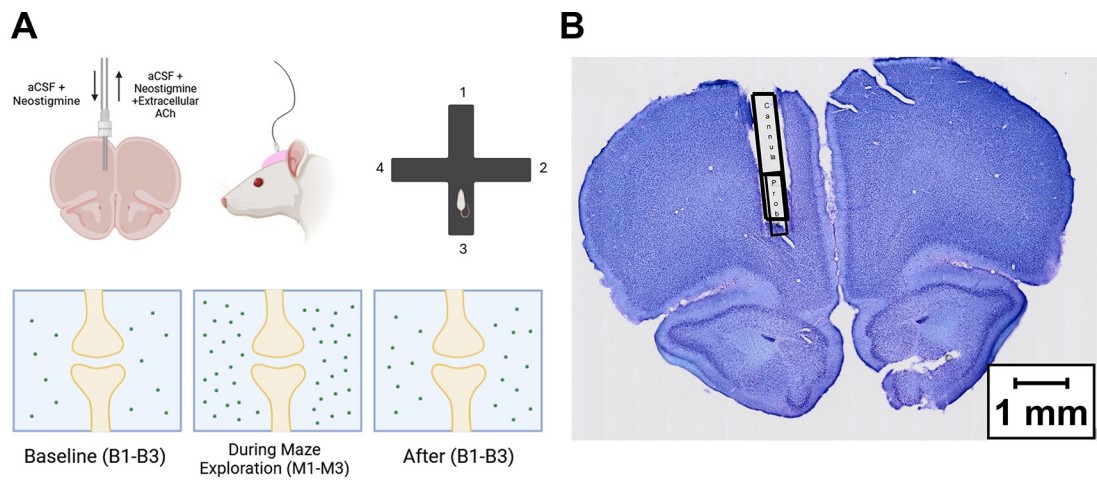

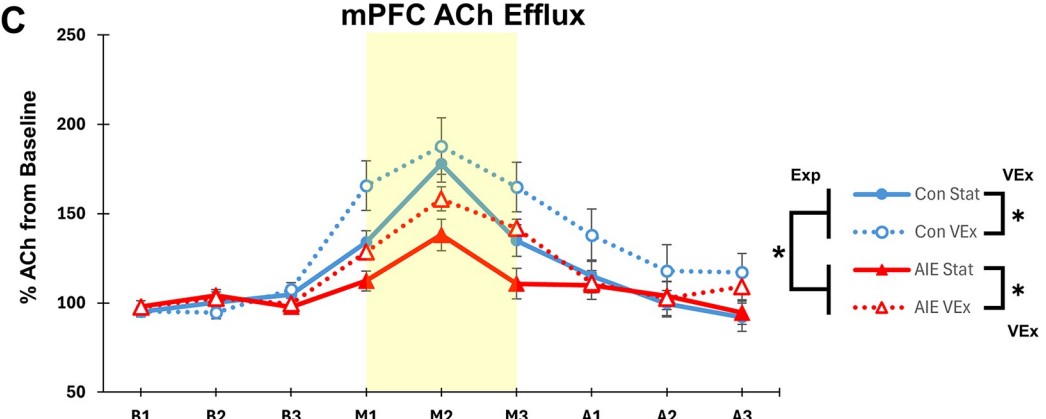

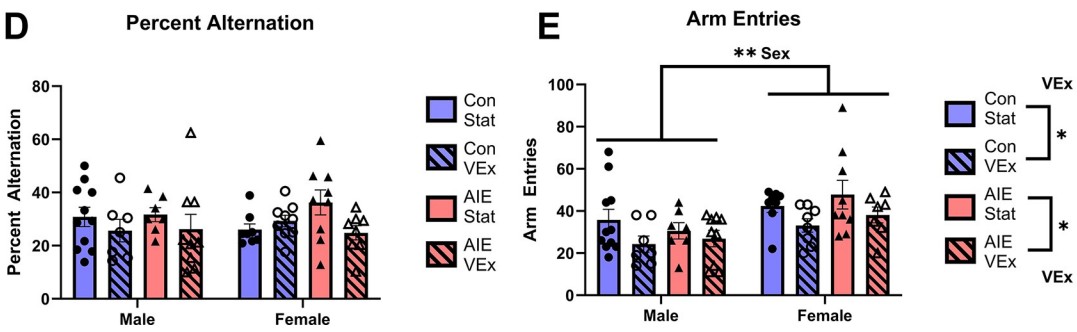

**Fig 3.** *In vivo* **microdialysis for ACh efflux during spontaneous alternation.** Schematic showing microdialysis setup, with guide cannulae implanted in the mPFC and neostigmine + aCSF infused to detect extracellular ACh efflux during four-arm spontaneous alternation maze exploration (**A**). Cresyl violet staining showing typical cannula placement in the mPFC (**B**). ACh efflux in the mPFC during spontaneous alternation (Males: Con-Stat = 11, Con-VEx = 7, AIE-Stat = 7, AIE-VEx = 9; Females: Con-Stat = 8, Con-VEx = 9, AIE-Stat = 9, AIE-VEx = 8) (**C**). All animals showed significant rises in ACh during maze exploration compared to baseline, but this rise was blunted in AIE animals and was improved in exercise animals, irrespective of AIE treatment. Percent alternation score during spontaneous alternation (**D**). Spontaneous alternation score did not differ as a function of AIE treatment or Exercise. Number of arm entries made during spontaneous alternation (Males: Con-Stat = 11, Con-VEx = 7, AIE-Stat = 7, AIE-VEx = 9; Females: Con-Stat = 8, Con-VEx = 8, AIE-Stat = 9, AIE-VEx = 7). Female rats made more arm entries than males irrespective of treatment group. Arm entries did not differ as a function of AIE, but VEx animals made less arm entries than stationary controls (**E**). $*p < 0.05$; $**p < 0.01$. Exp = Main effect of Exposure; VEx = Main Effect of Voluntary Wheel Exercise; Sex = Main effect of Sex.

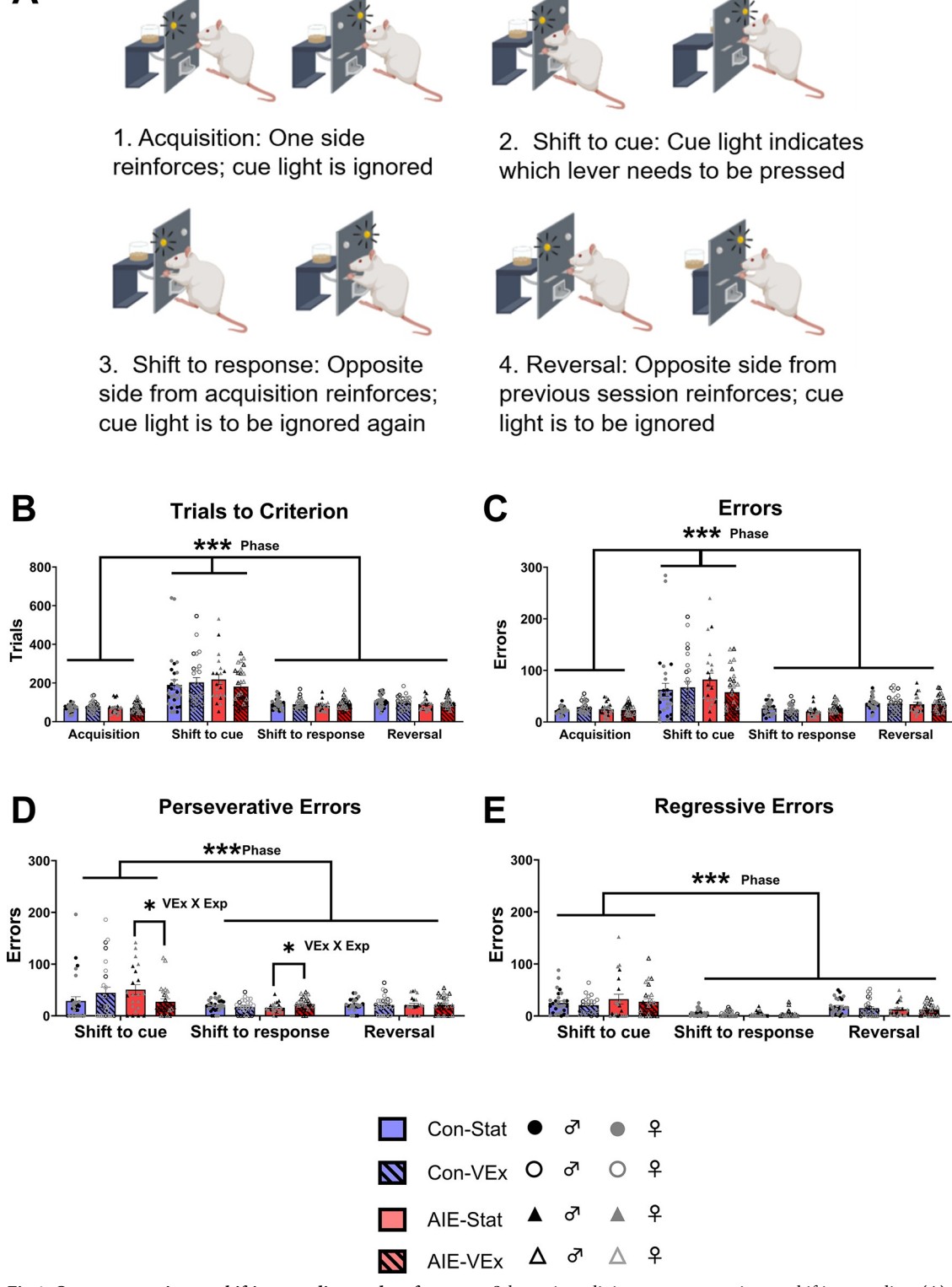

**Fig 4. Operant attention set shifting paradigm and performance.** Schematic outlining operant attention set shifting paradigm (**A**). Sessions 2, 3, and 4 included 20 trials with the previous day's rule, followed by a shift to a new rule. Criteria was 10 consecutive correct trials. Animals that did not reach criteria after 220 trials were rerun on the following day for 200 trials with the rule change in effect until they reached criterion. Trials to criterion on each phase of the task (Males: Con-Stat = 13, Con-VEx = 9, AIE-Stat = 9, AIE-VEx = 16; Females: Con-Stat = 17, Con-VEx = 17, AIE-Stat = 12, AIE-VEx = 11) (**B**). The number of trials needed to reach

criterion differed as a function of task phase, with the shift to cue phase being the most difficult for all subjects. However, trials to criterion did not differ based on Sex or Treatment group. Number of errors made on each phase of the task (Males: Con-Stat = 11, Con-VEx = 9, AIE-Stat = 9, AIE-VEx = 16; Females: Con-Stat = 17, Con-VEx = 17, AIE-Stat = 13, AIE-VEx = 10) **(C)**. The number of errors made differed as a function of task phase, with the shift to cue phase eliciting the most errors but was not affected by sex or treatment group. The number of perseverative errors made during the phases of the task that included a rule change (Males: Con-Stat = 12, Con-VEx = 9, AIE-Stat = 9, AIE-VEx = 15; Females: Con-Stat = 16, Con-VEx = 17, AIE-Stat = 13, AIE-VEx = 11) **(D)**. The number of perseverative errors changed as a function of phase, with the shift to cue eliciting the most errors. Additionally, there was a significant Session × AIE × VEx interaction, where Con-Stat and AIE-VEx animals made more perseverative errors on the shift to cue phase than Con-VEx and AIE-Stat, but less errors on the shift to response phase. The number of regressive errors made during the phases of the task that included a rule change (Males: Con-Stat = 12, Con-VEx = 9, AIE-Stat = 9, AIE-VEx = 15; Females: Con-Stat = 16, Con-VEx = 17, AIE-Stat = 13 AIE-VEx = 11) **(E)**. The number of regressive errors changed as a function of phase, with the shift to cue generating the most errors. However, it did not change as a function of sex or treatment group. \*\*\*p<0.001; \*p < 0.05. Phase = Main effect of task phase; VEx × Exp = Voluntary exercise by exposure by phase interaction.

Analysis of the latency to lever press (see Fig 5A) revealed a main effect of Task (F[2.176, 208.885] = 14.998, p < 0.001), as all rats got progressively quicker to lever press with each subsequent phase of the task. Post hoc analyses revealed that subjects were quicker on the reversal phase than acquisition and shift to cue (all p's < 0.05), while shift to response was significantly quicker than acquisition only (p < 0.05). However, there was no effect of Exposure or Sex on the latency to lever press nor were there any interactions (all p's > 0.32). Four subjects had their data excluded from this analysis as they were outliers on one or more phases (Final numbers in Fig 5).

Analysis of the latency to retrieve the sucrose pellet reward (see Fig 5B) following a correct choice revealed a main effect of Task (F[2.263, 208.240] = 5.574, p = 0.003), as subjects were quicker to collect the reward during the reversal phase. However, post hoc analyses revealed only a trending difference between the reversal phase and the shift to cue phase (p = 0.054), but no others. There was no effect of Exposure, Sex, or Exercise on the latency to collect reward nor were there any interactions (all p's > 0.08). Four subjects had their data excluded from this analysis as they were outliers on one or more phases (Final numbers in Fig 5).

### Experiment 2: Epigenetic changes in *Chat* genes

**ChIP analysis of H3K9me2 occupancy at the *Chat* promoter.** To determine if *Chat* gene expression is altered by an epigenetic mechanism following AIE in the male and female basal forebrain, we assessed histone methylation within the *Chat* promoter and *Chat* promoter CpG island. Assessment of adult basal forebrain levels of H3K9me2 occupancy at the *Chat* promoter revealed a significant interaction of Sex × Treatment (F[1, 66] = 14.06, p < 0.001; see Fig 6A). Follow-up comparison revealed that this effect was driven by increased occupancy of H3K9me2 at the *Chat* promoter of adult male AIE-treated animals in both the stationary (*p* < 0.001) and voluntary exercise conditions (*p* < 0.05). We did not observe changes in H3K9me2 occupancy at the Chat promoter in the adult female basal forebrain. Thus, AIE treatment led to long-lasting increases of H3K9me2 occupancy at the *Chat* promoter in adult males, but not females, an effect that was not influenced by exercise exposure in adulthood.

Assessment of basal forebrain levels of H3K9me2 occupancy at the *Chat* promoter CpG island revealed a significant interaction of Treatment × Exercise (F[1, 65] = 4.82, p < 0.05; see Fig 6B). Follow-up comparison revealed that regardless of sex, AIE treatment increased occupancy of H3K9me2 at the *Chat* promoter CpG island (Collapsed across Sex: *p* < 0.001), an effect that was reversed by exercise exposure in adulthood (Collapsed across Sex: *p* < 0.05). Thus, AIE treatment increased occupancy of H3K9me2 at the *Chat* promoter CpG island in the adult male and female basal forebrain that was reversed by exercise exposure in adulthood.

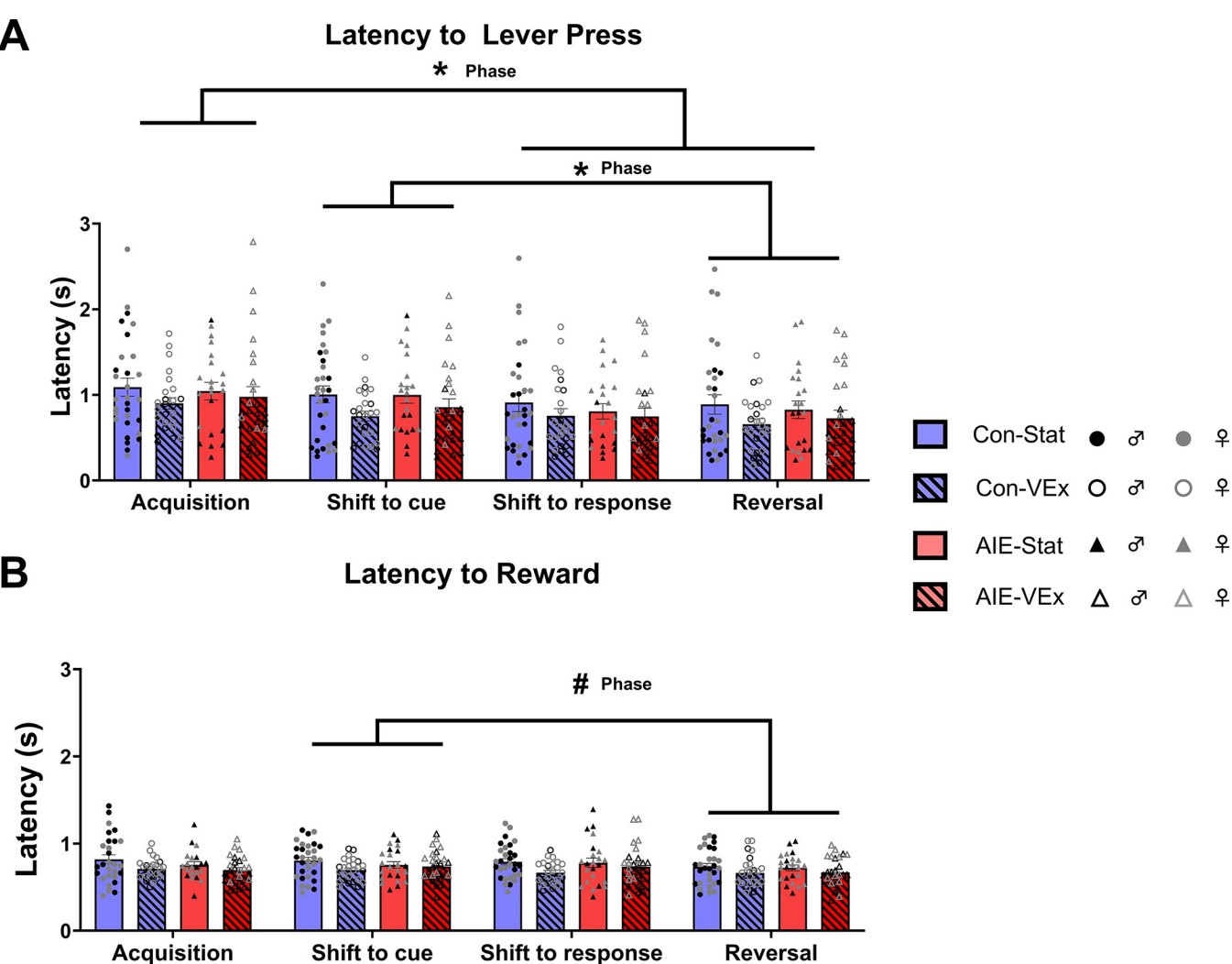

**Fig 5. Latency to lever press and retrieve reward during operant attention set shifting.** Overall latency to lever press during each phase (Males: Con-Stat = 13, Con-VEx = 9, AIE-Stat = 8, AIE-VEx = 14; Females: Con-Stat = 17, Con-VEx = 16, AIE-Stat = 14, AIE-VEx = 12) **(A)**. There was a significant effect of session, such that all animals became quicker to lever press during later sessions. Rats were significantly quicker during acquisition than shift to response and reversal, as well as quicker during shift to cue that during reversal. However, it did not change as a function of Sex or Treatment group. Latency to retrieve sucrose pellet following a correct choice (Males: Con-Stat = 12, Con-VEx = 9, AIE-Stat = 9, AIE-VEx = 15; Females: Con-Stat = 17, Con-VEx = 17, AIE-Stat = 13, AIE-VEx = 12) **(B)**. There was a significant effect of session, such that all animals became quicker to retrieve the sucrose pellet following a rule change during later sessions). However, post hoc analyses revealed that there was only a trending difference between the shift to cue and reversal phases. #p = 0.054, *p<0.05; **p<0.01. Phase = Main Effect of task phase.

## Discussion

Adolescence has been shown to be an incredibly important neurodevelopmental epoch, in which perturbations can lead to impairments that last into adulthood. Particularly, intermittent exposure to binge levels of ethanol intoxication has been shown to cause a nearly 30% reduction in ChAT expression throughout the basal forebrain [17, 18, 31, 41, 47], contributing to a loss of cholinergic tone in the cortex and impairment during cortical-dependent behavioral tasks [18]. However, it has been shown AIE does not lead to cell death in cholinergic cells, but rather the cholinergic phenotype is epigenetically silenced [31]. However, cholinergic phenotype in the MS/DB following AIE can be rescued using interventions such as cholinesterase inhibitors [47], anti-inflammatory drugs [41], and voluntary wheel running exercise

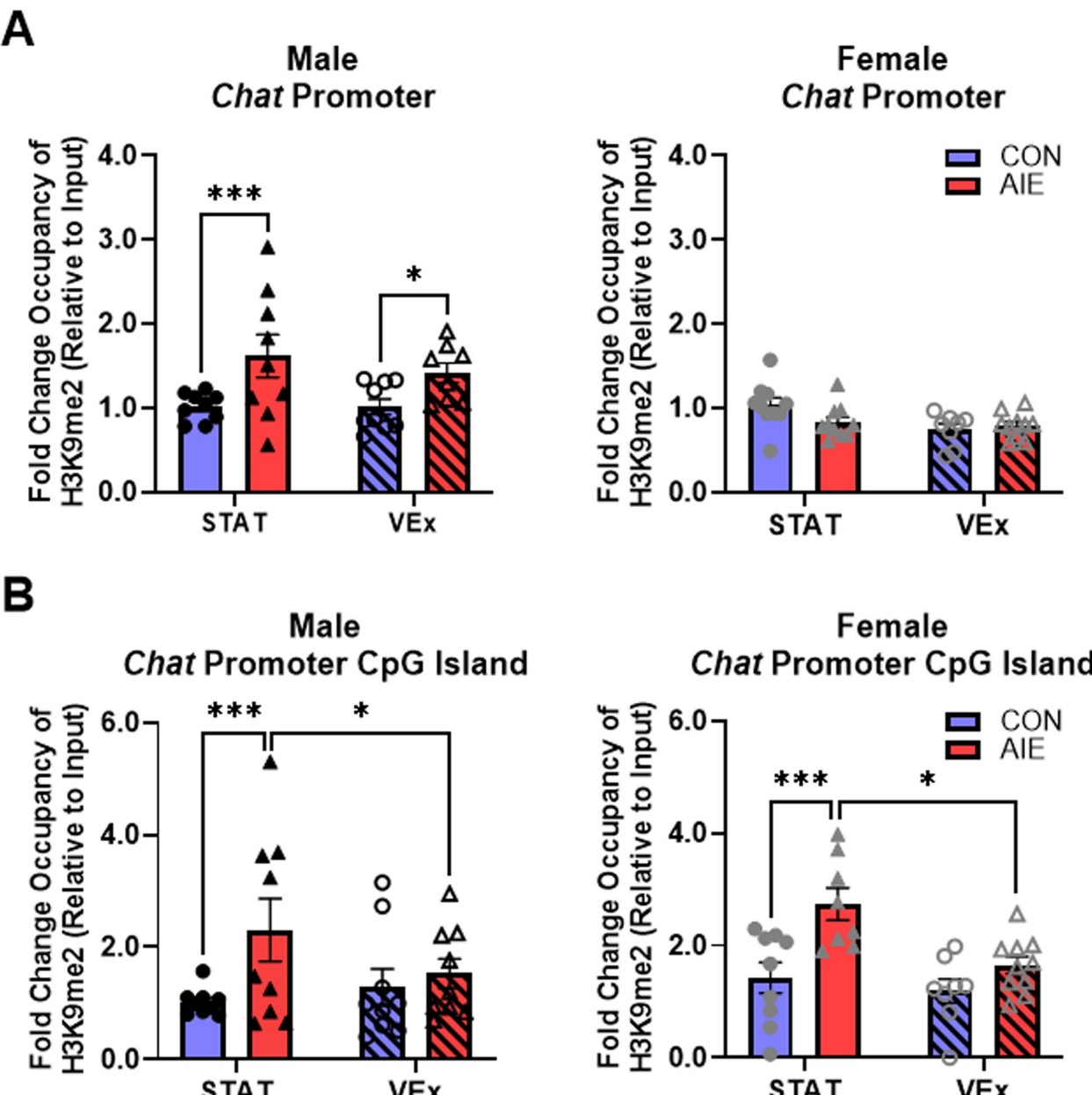

**Fig 6. Histone 3 lysine 9 dimethylation (H3K9me2) occupancy at *Chat* promoter regions.** (Males: Con-Stat = 10, Con-VEx = 10, AIE-Stat = 9, AIE-VEx = 10; Females: Con-Stat = 10, Con-VEx = 10, AIE-Stat = 10, AIE-VEx = 10). **(A)** Assessment of H3K9me2 occupancy at the *Chat* promoter revealed a significant interaction of Sex × Treatment, such that occupancy of H3K9me2 at the *Chat* promoter was increased in adult male AIE-treated animals in the stationary and voluntary exercise conditions. We did not observe changes in H3K9me2 occupancy at the *Chat* promoter in the adult female basal forebrain. **(B)** Assessment of basal forebrain levels of H3K9me2 occupancy at the *Chat* promoter CpG island revealed a significant interaction of Treatment × Exercise such that that regardless of sex, AIE treatment increased occupancy of H3K9me2 at the *Chat* promoter CpG island, an effect that was reversed by exercise exposure in adulthood. Data are presented as mean ± SEM. ChIP was run in duplicate. * p < 0.05, *** p < 0.001.

[31, 41]. In particular, voluntary wheel running increases neurotrophins, both BDNF and NGF, and NGF has been shown to be responsible for exercise-induced rescue of the cholinergic phenotype following alcohol-related brain damage [44, 45]. Thus, exercise represents an intriguing potentiality to rescue some of the downstream consequences of cholinergic silencing following AIE, particularly in the prefrontal cortex.

The main findings of the current study are that AIE leads to a blunting of behaviorally evoked ACh efflux in the mPFC during spontaneous alternation, which was rescued with exercise in both males and females. It should be noted that exercise also increased ACh efflux in control rats, demonstrating that exercise has a broader action to improve brain function. Furthermore, AIE causes an upregulation of histone methylation marker H3K9me2 at the *Chat* promoter CpG island in the basal forebrain of male and female rats, which was corrected by exercise. Increased H3K9me2 and DNA methylation on the CpG island are associated with stable transcriptional gene silencing [31], suggesting that AIE lead to an epigenetic silencing of the ChAT gene leading to suppression of the ChAT phenotype and ACh—which are recovered with exercise. The ability of VEx to rescue mPFC ACh efflux and epigenetic repression of the cholinergic phenotype (i.e., *Chat* promoter CpG island) following AIE supports the hypothesis that ChAT+ cells of the basal forebrain enter into a quiescent state, but their phenotype and function can be restored. The current data complements previous work showing that exercise [31, 41], cholinesterase inhibiting drugs [47], and modulation of the p75NTR [41] are sufficient to rescue the cholinergic phenotype in the basal forebrain.

This is the first study that examined epigenetic changes in the cholinergic phenotype following AIE in both male and female rodents. We did find a sex-specific effect: The increased occupancy of H3K9me2 at the *Chat* promoter was only seen in adult male AIE-treated and was not corrected by exercise. This is contrasted with the sex-independent effect of the upregulation of histone methylation marker H3K9me2 at the more specific region of the *Chat* promoter CpG island that controls gene silencing, which was corrected by exercise. While it had previously been known that this capacity exists in the septohippocampal circuit [45], it was unknown whether exercise-induced plasticity was able to rescue the cholinergic projections from the NbM/SI complex to the prefrontal cortex. The current data supports that hypothesis and suggests that both cholinergic circuits in the basal forebrain display a similar capacity for restoration with exercise.

There were also sex specific effects in weight gain during AIE treatment. The effect that AIE-induced suppression weight gain (less than 10%) selectively in males during gavage treatment was not expected, but we did observe a similar phenomenon in a recently published study that also included a larger sample size [50]. This sex differences may be due to differences in ethanol metabolism between the sexes–adult females metabolize ethanol quicker than adult males [51] and may have a briefer intoxication period that could lead to the resumption of typical food consumption in females. However, other work that used a chronic ip injection of ethanol across adolescence found that both male and female rats chronically exposed to ethanol gained less weight gain than saline exposed rats [52]

The mechanism for the exercise-induced recovery of cholinergic tone is likely threefold: One, voluntary wheel running exercise mirrors the effects of anti-inflammatory drugs such as indomethacin following AIE [41]. In particular, exercise can inhibit pro-inflammatory signaling cascades and reverse epigenetic silencing of the ChAT promoter region [38]. Therefore, through a reduction in inflammatory signaling, exercise contributes to a restoration of the cholinergic phenotype. Secondly, exercise increases the content of neurotrophins, particularly NGF and BDNF [44, 45]. AIE has been shown to lead to reductions in these neurotrophins [50], while exercise induced recovery of the cholinergic phenotype following pyrithiamine deficiency is dependent on NGF [44]. Such an enhancement in neurotrophins likely explains the increase in ACh efflux in control rats following exercise, as exercise has been shown to increase both BDNF and NGF in both brain damaged and normal rodents [45]. Lastly, modulation of the p75NTR, which modulates ChAT expression genes, has been shown to restore the cholinergic phenotype following AIE [41], which represents another potential mechanism by which exercise exerts its effects.

The current spontaneous alternation data mirrors previous work assessing spatial working memory following AIE, as it has been shown that this model does not affect performance in young adult rats on either spontaneous alternation [19] or acquisition of spatial location in the Barnes maze or Morris water maze [18, 53, 54]. While learning initial spatial associations are not impaired AIE, once spatial associations are learned, AIE impairs the ability to update these associations with new contingencies, as reflected by reversal learning [31, 53–55].

The evidence regarding the modulation of spatial working memory following exercise is a bit more mixed. It has been shown previously that wheel running exercise is enough to rescue spontaneous alternation behavior following pyrithiamine deficiency, which imparts much more severe brain pathology than AIE [45]. Surprisingly, there is data to suggest that exercise may even impair spontaneous alternation behavior on a three-arm maze, even though it leads to an increase in hippocampal neurogenesis [56]. Meanwhile, other literature demonstrates an improvement of spatial working memory by exercise on the Morris water maze, which is due to an increase in BDNF [57]. One possible reason for this discrepancy is that it is possible that different intensities of exercise impart differential benefits, as it has been shown that lower intensities of 30 minute forced treadmill running over the course of 30 days to benefit short term delayed working memory, while higher intensities benefit long-term working memory [58]. Another issue is task complexity or difficulty.

In contrast with previous studies [18, 28], AIE deficits in cognitive flexibility, as measured by the total number of total errors, during ASST were not observed. One of the main differences between the methodology of the current study and the previous study is that the rule change was introduced after 20 trials into the session in the current study, while the previous study introduced the rule change at the beginning of each session. This may contribute to some of the differences seen between the results of this study and previous ones, as the difficulty of the task may be masking some potential differences between AIE and control animals. A previous study where AIE treated animals shifted from cue light to response only showed an increase in regressive errors exclusively in male rats, but not trials to criterion, [29] which contrasts with the data presented here that suggests that AIE animals make more perseverative errors during the shift to cue, but less during the shift to side phase. Therefore, these deficits may differ based on task demand; AIE animals do show an increased number of total errors during extradimensional set shifting when presented with a series of reversals on prior sessions [55], suggesting that AIE impairs cognitive flexibility only selectively on certain set shifting tasks, and potentially interacting with exercise to affect perseverative errors during extradimensional shifts.

The fact that the current study did not find AIE-induced behavioral flexibility deficits during reversal learning in the operant paradigm is congruent with what we've reported using this task in a past study from our lab [18]. While there is evidence to suggest that AIE does impact reversal learning [28, 30, 59], it has not been shown using the operant attention set shifting paradigm used in the current study. Previous data show that reversal learning deficits on the Morris water maze are sex dependent as well, with males impaired during a short-term relearning test [60], suggesting that the severity of reversal learning deficits change as a function of factors such as sex. Since AIE leads to significant cholinergic pathology in the OFC such as a reduction in vesicular acetylcholine transporter and a decrease in behaviorally evoked ACh efflux [42], it is possible that AIE impairs some, but not other behaviors dependent on the OFC. Previous data shows that AIE reduces functional connectivity between structures such the nucleus accumbens and hippocampus, as well as the prelimbic cortex and thalamus, but interestingly, none with the OFC [55]. There is also data to suggest that the lateral habenula [61] and the ventrolateral striatum [62] are regions important to the regulation of switching strategies, and therefore this particular task may not be sensitive to AIE induced deficits to the OFC, but rather depend on a more diffuse set of structures.

Overall, the data from the current study supports the hypothesis that the cholinergic system, specifically both critical forebrain projection systems, remains plastic into adulthood following alcohol-related pathology brain induced during adolescence. Furthermore, by the inclusion of both females and males we demonstrate the exercise is a robust remediation to rescue ACh efflux impairments in the mPFC caused by AIE, as well as enhancement of ACh efflux in adult control rats. Furthermore, exercise selectively corrects the epigenetic machinery that leads to silencing of cholinergic genes following AIE. However, we did not see compelling AIE or exercise effects on behavior suggesting that lasting cognitive flexibility deficits caused by AIE are likely modulated by task demand, as previous data would suggest for behavioral flexibility. Thus, future work is needed to discern the unique components of the attentional and cognitive domains that are affected by chronic adolescent alcohol exposure. Nevertheless, these data overall support the utility of exercise in rescuing frontocortical alcohol related brain damage caused during adolescence and demonstrate the importance of brain plasticity in recovery.

## Supporting information

**S1 Fig. Operant attention set shifting performance for males.** Trials to criterion on each phase of the task (Con-Stat = 13, Con-VEx = 9, AIE-Stat = 9, AIE-VEx = 16) **(A)**. Number of errors made on each phase of the task (Males: Con-Stat = 11, Con-VEx = 9, AIE-Stat = 9, AIE-VEx = 16) **(B)**. The number of perseverative errors made during the phases of the task that included a rule change (Con-Stat = 12, Con-VEx = 9, AIE-Stat = 9, AIE-VEx = 15) **(C)**. The number of regressive errors made during the phases of the task that included a rule change (Con-Stat = 12, Con-VEx = 9, AIE-Stat = 9, AIE-VEx = 15) **(D)**. Overall latency to lever press during each phase (Con-Stat = 13, Con-VEx = 9, AIE-Stat = 8, AIE-VEx = 14) **(E)**. Latency to retrieve sucrose pellet following a correct choice (Con-Stat = 12, Con-VEx = 9, AIE-Stat = 9, AIE-VEx = 15) **(F)**.
(TIF)

**S2 Fig. Operant attention set shifting performance for females.** Trials to criterion on each phase of the task (Con-Stat = 17, Con-VEx = 17, AIE-Stat = 12, AIE-VEx = 11) **(A)**. Number of errors made on each phase of the task (Con-Stat = 17, Con-VEx = 17, AIE-Stat = 13, AIE-VEx = 10) **(B)**. The number of perseverative errors made during the phases of the task that included a rule change (Con-Stat = 16, Con-VEx = 17, AIE-Stat = 13, AIE-VEx = 11) **(C)**. The number of regressive errors made during the phases of the task that included a rule change (Con-Stat = 16, Con-VEx = 17, AIE-Stat = 13 AIE-VEx = 11) **(D)**. Overall latency to lever press during each phase (Con-Stat = 17, Con-VEx = 16, AIE-Stat = 14, AIE-VEx = 12) **(E)**. Latency to retrieve sucrose pellet following a correct choice (Con-Stat = 17, Con-VEx = 17, AIE-Stat = 13, AIE-VEx = 12) **(F)**.
(TIF)

**S1 File. Raw data for experiments 1 and 2.** Raw data for all analyses, with subjects identified by ID number, sex, litter, gavage treatment, and exercise condition.
(PDF)

## Author Contributions

**Conceptualization:** Matthew J. Fecik, Polliana T. Nunes, Ryan P. Vetreno, Lisa M. Savage.

**Data curation:** Matthew J. Fecik, Polliana T. Nunes, Ryan P. Vetreno.

**Formal analysis:** Matthew J. Fecik.

**Funding acquisition:** Lisa M. Savage.

**Investigation:** Matthew J. Fecik, Ryan P. Vetreno.

**Methodology:** Matthew J. Fecik, Polliana T. Nunes, Ryan P. Vetreno, Lisa M. Savage.

**Supervision:** Lisa M. Savage.

**Validation:** Matthew J. Fecik, Polliana T. Nunes, Ryan P. Vetreno.

**Visualization:** Matthew J. Fecik, Ryan P. Vetreno.

**Writing – original draft:** Matthew J. Fecik, Ryan P. Vetreno.

**Writing – review & editing:** Polliana T. Nunes, Lisa M. Savage.

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
