## [Decision Letter · Decision Letter 0]

23 Jul 2024

PONE-D-24-25335Voluntary wheel running exercise rescues behaviorally-evoked acetylcholine efflux in the medial prefrontal cortex and epigenetic changes in ChAT genes following adolescent intermittent ethanol exposurePLOS ONE

Dear Dr. Savage,

Thank you for submitting your manuscript to PLOS ONE. After careful consideration, we feel that it has merit but does not fully meet PLOS ONE’s publication criteria as it currently stands. Therefore, we invite you to submit a revised version of the manuscript that addresses the points raised during the review process.

The reviewers' request significant modifications to the submitted manuscript. The methodology  and the results description should be more detailed and precise. Please modify these sections clarifying the reviewers' doubts and providing justification for their concerns.  There are inconsistencies between the abstract and the results section. Please also note that there are concerns regarding the analysis and presentation of sex differences, as well as the possibility of litter effects.

We look forward to receiving your revised manuscript.

Kind regards,

Yael Abreu-Villaça, Ph.D.

Academic Editor

PLOS ONE

Journal Requirements:

   "This work was funded by U01AA028710"

Reviewers' comments:

Reviewer's Responses to Questions

**Comments to the Author**

1. Is the manuscript technically sound, and do the data support the conclusions?

Reviewer #1: Partly

Reviewer #2: Partly

2. Has the statistical analysis been performed appropriately and rigorously? 

Reviewer #1: Yes

Reviewer #2: Yes

3. Have the authors made all data underlying the findings in their manuscript fully available?

Reviewer #1: Yes

Reviewer #2: Yes

4. Is the manuscript presented in an intelligible fashion and written in standard English?

Reviewer #1: Yes

Reviewer #2: Yes

5. Review Comments to the Author

Reviewer #1: The authors show that wheel running is able to reverse both behavioral and epigenetic deficits caused by adolescent alcohol exposure (repeated gavage). Together, these build on the observation that alcohol causes cognitive deficits due to cholinergic disfunction in the mPFC, and provide rationale for exercise as a therapeutic intervention to reverse alcohol effects (and perhaps provide a mechanism more broadly for exercise to restore cholinergic function and cognition across conditions). Indeed, wheel running increases mPFC cholinergic efflux in non-alcohol mice as well as AIE.

While the data are interesting, there are several important areas where the description of the data (e.g. in the abstract) seem not to conform to what the data actually shows, as well as other issues of presentation and clarity in the data (and especially showing significance in several figures). Some parts of the manuscript are specifically noted below, but similar changes should be made throughout.

One concern is in Fig.6, where AIE increases Chat markers in males, but wheel running does not reverse this effect. This would contradict the abstract that “VEx rescued the AIE induced deficits in … epigenetic changes in ChAT genes” (line35-36, also lines 94-96 and elsewhere in manuscript). The findings also contradict this sentence since it says “in both sexes” since Fig6 shows a wheel running reversal effect in females but not males. It is unclear from the text in lines 390 and on whether there are effects in males, which are not shown in the figure. In fact, lines 398-399 to say that wheel running “insignificantly blunted” the measure is not appropriate to say (especially in relation to how it is framed in the abstract and end of introduction, and in lines 408-409). That is, to say “exercise exposure in adulthood rescued H3K9me2 methylation of the Chat promoter CpG island across males and females” is not correct.

Another concern is in Figure 4D, where it is unclear that AIE itself causes changes in cognitive function. Lines 364 to 367 describe that there is an interactive effect, but this is not shown in the Figure, also lines 468- mention that AIE does not alter flexibility in the present model.

Also, in this section, there is no mention of testing for a sex effect (while line377 shows such a test for sex for a different measure), and this is the only measure with an AIE x wheel running effect.

In addition, since there are clear sex differences in the ChAT changes, the key changes in the behaviors in Fig4D should be shown separated by sex.

A further concern regarding cholinergic measures is that some references are described as showing that wheel running reverses AIE changes in epigenetic markers. Thus, it should be made explicit what is new about the measures performed here. Even if they are replications, that is fine, it should just be made clearer.

Thus, the manuscript overall needs much clearer description of the findings, especially the abstract, to note sex differences in observed patterns (as well as similarities between males and females). It should make clear the cases where AIE did had an effect on particular measures that was reversed by wheel running, with no such effect in controls. The manuscript and abstract should also be clearer where wheel running has an overall effect compared with specific effects in AIE. This occurs e.g. for cholinergic dialysis which is increased in both AIE and controls (lines 330 on, where there was no exposure exercise interaction), and lines 426-427 which only describe effects in AIE and doesn’t mention running enhancing ACh also in controls. These are important because it suggests that some effects of wheel running are surely valuable therapeutically, but are not simply “reversing” AIE effects.

In Fig3, the bars to compare significance across groups in the legend is confusing and should be made clearer. Similar in Fig4C,D for bars above the figures.

Reviewer #2: This manuscript addresses a relevant issue regarding long-term deleterious effect of ethanol exposure during adolescence upon cholinergic signaling and cognitive functions. Additionally, they employed voluntary activity later in life and investigated whether it might cause beneficial outcomes on ethanol-induced effects in their model. The study is strengthened by the inclusion of animals of both sexes as well as measures during the ethanol exposure period and at the time-point of behavioral evaluations. Although the experiments are pretty straight forward, there are some methodology, analysis and interpretive concerns. In the ensuing paragraphs these major and secondary concerns are described.

Major concerns:

1 - To this reviewer, a matter of concern is the number of animals used in each experimental condition. The authors stated that, for both experiments, “no more than two pups of each sex per litter was assigned to an experimental condition”. This brings a concern about litter-effect on this manuscript’s data. Firstly, how many litters were used in each experiment ?

This is an important issue since the variability within litters, as well-known, is lower than between. Thus, it is a major source of variability in an experiment and controlling it is crucial to enhance rigor and reproducibility. Thus, to avoid litter-effect on results the authors should use the mean of pups from each litter for each experimental condition. Accordingly, the sample size described in the manuscript should be corrected to present the adjusted numbers.

This brings another concern about whether the number of animals used was appropriated. In this sense, it is possible that some sample sizes are around 6 if using the mean per litter, which would be too small, particularly regarding behavioral evaluation.

To this reviewer it is critical that these questions are answered and that the answers reflect a reasonable experimental design.

2 – I understood that in experiment 1 mice were submitted to both behavioral tests. If that is correct, it is not clear why the sample sizes for the attention set shifting and reversal learning is bigger than the ones in microdialysis and spontaneous alternation. Please, clarify.

3 – It is not clear how animals used in the HPLC and cresyl violet staining were euthanized. Was it immediately after the attention set shifting and reversal learning test? What was the method used? Also, in line 243, it is described that in the experiment 1, all rats were anaesthetized and perfused. Is that correct? Please, provide a detailed description for euthanasia in experiment 1.

4 – In some points of the manuscript the authors give the idea that previous data had already shown that AIE model does not affect spontaneous alternation (lines 451-454). So, why did the authors used this test? Shouldn’t this be addressed as a limitation of the study?

5 – It is not clear to this reviewer why the authors presented the ChIP results separated by sex. In the statistical analysis, in the methods session, the authors stated that they used a 2 x 2 ANOVA. Thus, I assumed that sex was not a factor in this analysis. If that is correct, the authors must provide rationale for, a priori, analyze males and females separately only for this variable. Otherwise, they should perform the analysis including sex as a factor. Please clarify.

Minor:

1 – The description of the attention set shifting and reversal learning test could be improved if reorganized. For instance, in the second paragraph describes the acquisition session, which involves a response based on animal’s preferred side. The establishment of the side preference was not described yet. As a suggestion, considering the complexity of the protocol, the authors could describe it chronologically.

2 – The lines 269-272 should be revised.

3 – Some results are not addressed in the discussion session. For instance, was the sex difference in weight gain expected or not? What could explain it? Additionally, there was an effect of Exercise in the levels of H3K9me2 at Chat promoter of females (but, please, see comment 4 in major about the analysis of this variable). However, it is not discussed. Could it be related to the hyperactivity in females compared to males?

4 – The conclusion session should be revised. The authors wrote that: “… AIE, its effects on behaviors modulated by the cholinergic system are variable.”. It is difficult to conclude this since there are limitations in the behavioral testes used in this work (lines 451-454; 474-475; 488-495). Besides that, afterwards, the authors address phasic and tonic ACh signaling and attention set shifting behaviors. It was not addressed anywhere else in the manuscript. It is difficult to understand the impact/importance of these types of signaling to the study. If it is important, please provide rationale for it in the introduction and discussion sessions.

6. PLOS authors have the option to publish the peer review history of their article (what does this mean?). If published, this will include your full peer review and any attached files.

Reviewer #1: No

Reviewer #2: No

---

## [Author Response · Author response to Decision Letter 0]

27 Aug 2024

August 26, 2024

Dear Dr. Abreu-Villaça

Enclosed is the revised manuscript entitled “Voluntary wheel running exercise rescues behaviorally -evoked acetylcholine efflux in the medial prefrontal cortex and epigenetic changes in ChAT genes following adolescent intermittent ethanol exposure " for consideration for publication as a research article in Plos One.

We want to thank the Reviewers for their helpful comments, which have improved the clarity of the manuscript. All changes are marked by red text. A point-by-point summary of the Reviewers’ comments and our responses is included. We believe that we have addressed all their concerns. 

Thank you for considering this manuscript for publication in Plos One.

Sincerely,

Lisa M. Savage, Ph.D

Professor of Behavioral Neuroscience

Binghamton University, State University of New York

PONE-D-24-25335: 

Reviewer #1: 

1. One concern is in Fig.6, where AIE increases Chat markers in males, but wheel running does not reverse this effect. This would contradict the abstract that “VEx rescued the AIE induced deficits in … epigenetic changes in ChAT genes” (line35-36, also lines 94-96 and elsewhere in manuscript). The findings also contradict this sentence since it says “in both sexes” since Fig 6 shows a wheel running reversal effect in females but not males. 

Response: We thank the reviewer for this comment. As Reviewer 2 requested inclusion of Sex as a factor in analysis of ChIP data, we have revised our conclusions to reflect these changes. Specifically, in the Abstract, we now report, “It was found that VEx rescued the AIE induced deficits in mPFC ACh efflux and epigenetic methylation at the Chat promoter CpG island across sexes.”

2. It is unclear from the text in lines 390 and on whether there are effects in males, which are not shown in the figure. In fact, lines 398-399 to say that wheel running “insignificantly blunted” the measure is not appropriate to say (especially in relation to how it is framed in the abstract and end of introduction, and in lines 408-409). That is, to say “exercise exposure in adulthood rescued H3K9me2 methylation of the Chat promoter CpG island across males and females” is not correct.

Response: We now include revised data reflecting inclusion of Sex as a factor in the ChIP analysis and the section was rewritten.

3. Another concern is in Figure 4D, where it is unclear that AIE itself causes changes in cognitive function. Lines 364 to 367 describe that there is an interactive effect, but this is not shown in the Figure, also lines 468- mention that AIE does not alter flexibility in the present model. Also, in this section, there is no mention of testing for a sex effect (while line377 shows such a test for sex for a different measure), and this is the only measure with an AIE x wheel running effect. In addition, since there are clear sex differences in the ChAT changes, the key changes in the behaviors in Fig4D should be shown separated by sex.

Response: We have amended the figure to more clearly demonstrate the interaction. Additionally we have clarified in the discussion: “In contrast with previous studies [18, 28], AIE deficits in cognitive flexibility, as measured by the total number of total errors, during ASST were not observed.” to demonstrate that we are referring to total errors, while there were potential deficits in perseverative errors. The following was added to the discussion to address this result in particular: “A previous study where AIE treated animals shifted from cue light to response only showed an increase in regressive errors exclusively in male rats, but not trials to criterion, [29] which contrasts with the data presented here that suggests that AIE animals make more perseverative errors during the shift to cue, but less during the shift to side phase. Therefore, these deficits may differ based on task demand; AIE animals do show an increased number of total errors during extradimensional set shifting when presented with a series of reversals on prior sessions [55], suggesting that AIE impairs cognitive flexibility only selectively on certain set shifting tasks, and potentially interacting with exercise to affect perseverative errors during extradimensional shifts”.

As for the testing of a sex effect on the number of perseverative and regressive errors, we have now clarified in the results that Sex was included in the analysis but was not significant. Additionally, we have added supplemental figures of the operant data for each sex (Males: Supplementary Figure 1; Females: Supplementary Figure 2).

4. A further concern regarding cholinergic measures is that some references are described as showing that wheel running reverses AIE changes in epigenetic markers. Thus, it should be made explicit what is new about the measures performed here. Even if they are replications, that is fine, it should just be made clearer.

Response: On page 20 we now discuss that this is the first study that used both sexes to examine epigenetic changes at the Chat promotors in the NbM region following AIE. The data replicate the male data previously reported in the MS/DB region (31,41).

5. Thus, the manuscript overall needs much clearer description of the findings, especially the abstract, to note sex differences in observed patterns (as well as similarities between males and females). It should make clear the cases where AIE did had an effect on particular measures that was reversed by wheel running, with no such effect in controls. The manuscript and abstract should also be clearer where wheel running has an overall effect compared with specific effects in AIE. This occurs e.g. for cholinergic dialysis which is increased in both AIE and controls (lines 330 on, where there was no exposure exercise interaction), and lines 426-427 which only describe effects in AIE and doesn’t mention running enhancing ACh also in controls. These are important because it suggests that some effects of wheel running are surely valuable therapeutically, but are not simply “reversing” AIE effects.

Response: We now state more clearly the Sex effects (see 1 above) in all analyses. In addition, we now also make it clear that exercise increases mPFC ACh in both control and AIE rats (see pages 19 and 21), demonstrating the broad effects of exercise.

6. In Fig3, the bars to compare significance across groups in the legend is confusing and should be made clearer. Similar in Fig4C,D for bars above the figures.

Response: We thank the reviewer for this comment. We have altered the bars on the figures to be clearer and have additionally altered the figure legends as well.

Reviewer #2: 

1. To this reviewer, a matter of concern is the number of animals used in each experimental condition. The authors stated that, for both experiments, “no more than two pups of each sex per litter was assigned to an experimental condition”. This brings a concern about litter-effect on this manuscript’s data. Firstly, how many litters in each experiment were used? 

Response: We have since run an analysis of all of our major behavioral variables (and HPLC) from Experiment 1 with litter as a factor and did not find any significant differences based on this variable. In addition, we have included the total number of litters used for each experiment. The following text was included: “A total of 17 litters were used in this experiment and in most cases only 1 pup per sex per litter was assigned to an experimental condition (Sex, Exposure, Exercise); However, in some cases 2 pups of each sex per litter were assigned to an experimental condition. In the interest of controlling for potential litter effects, litter was included as a factor in analyses of all behavioral and microdialysis data. There were no litter effects nor any interactions involving litter on any of these measures (all p’s > 0.05), and therefore data are represented here without litter as a factor”

The following text was included for Experiment 2: “A total of 10 litters for Experiment 2 were produced with most commonly 1 pup per sex of each litter assigned to an experimental condition, but in some instances 2 pups per sex/litter were assigned to an experimental condition.”

2. I understood that in experiment 1 were submitted to both behavioral tests. If that is correct, it is not clear why the sample size for the attention set shifting and reversal learning is bigger than in microdialysis and spontaneous alternation. Please, clarify it to the reader.

Response: The reason that the sample size is bigger for attention set shifting is because we excluded animals with non-detectable ACh peaks and misplaced cannulae from our analysis of spontaneous alternation and HPLC. This is now clarified in the methods section with the following text: “Only animals with detectable ACh levels and accurate cannula placement were included in spontaneous alternation and microdialysis data analysis (Males: Con-Stat = 11, Con-VEx = 7, AIE-Stat = 8, AIE-VEx = 9; Females: Con-Stat = 9, Con-VEx = 9, AIE-Stat = 11, AIE-VEx = 8)”

3. It is not clear how animals used in the HPLC and cresyl violet staining were euthanized. Was it immediately after the attention set shifting and reversal learning test? What was the method used? Also, in line 243, it is described that, in the experiment 1, all rats were anaesthetized and perfused. Is that correct? Please, provide a detailed description for euthanasia in experiment 1.

Response: We have now clarified that all the animals from Experiment 1 were perfused following attention set shifting. 

4. In some points of the manuscript the authors give the idea that previous data already shown that AIE model does not affect spontaneous alternation (lines 451-454). So, why did the authors used this test? Shouldn’t this be addressed as a limitation of the study?

Response: On page 8 we now state that spontaneous alternation is a task that consistently evokes activity/cognitive dependent release of ACh in the frontal cortex and hippocampus. Thus, it serves as an assay of activity-dependent release of ACh.

5. It is not clear to this reviewer why the authors presented the ChIP results separated by sex. In the statistical analysis, in the methods session, the authors stated that they used a 2 x 2 ANOVA. Thus, I assumed that sex was not a factor in this analysis. If that is corrected, the authors must provide rationale for, a priori, analyze males and females separately only for this variable. Otherwise, they should perform the analysis including sex as a factor. Please clarify it.

Response: We thank the Reviewer for this comment and agree that inclusion of Sex as a factor is critical. In the revision, we conducted 2 � 2 � 2 ANOVAs with follow-up analyses performed using Bonferroni corrections for simple main effect comparisons. We have revised the Results pertaining to the ChIP data as follows, “To determine if Chat gene expression is altered by an epigenetic mechanism following AIE in the male and female basal forebrain, we assessed histone methylation within the Chat promoter and Chat promoter CpG island. Assessment of adult basal forebrain levels of H3K9me2 occupancy at the Chat promoter revealed a significant interaction of Sex × Treatment (F[1, 66] = 14.06, p < 0.001) (see Figure 6A). Follow-up comparison of simple main effects revealed that this effect was driven by increased occupancy of H3K9me2 at the Chat promoter of adult male AIE-treated animals in the stationary (Bonferroni adjusted: p < 0.001) and voluntary exercise conditions (Bonferroni adjusted: p < 0.05). We did not observe changes in H3K9me2 occupancy at the Chat promoter in the adult female basal forebrain. Thus, AIE treatment led to long-lasting increases of H3K9me2 occupancy at the Chat promoter in adult males, but not females that was unaffected by exercise exposure in adulthood. 

Assessment of basal forebrain levels of H3K9me2 occupancy at the Chat promoter CpG island revealed a significant interaction of Treatment × Exercise (F[1, 65] = 4.82, p < 0.05) (see Figure 6B). Follow-up comparison of simple main effects revealed that regardless of sex, AIE treatment increased occupancy of H3K9me2 at the Chat promoter CpG island (Collapsed across Sex: Bonferroni adjusted: p < 0.001), an effect that was reversed by exercise exposure in adulthood (Collapsed across Sex: Bonferroni adjusted: p < 0.05). Thus, AIE treatment increased occupancy of H3K9me2 at the Chat promoter CpG island in the adult male and female basal forebrain that was reversed by exercise exposure in adulthood.

6. The description of the attention set shifting and reversal learning test could be improved if reorganized. For instance, in the second paragraph describes the acquisition session, which involves a response based on animal’s preferred side. The establishment of the side preference was not described yet. As a suggestion, considering the complexity of the protocol, the authors could describe it chronologically. 

Response: We have removed the paragraph regarding the attention set shifting task that precedes the paragraph on pretraining. All of the information from that paragraph is addressed subsequently and therefore was redundant and negatively impacted understandability.

7. The lines 269-272 should be revised.

Response: This section was rewritten.

8. Some results are not addressed in the discussion session. For instance, was the sex difference in weight gain expected or not? What could explain it? Additionally, there was an effect of Exercise in the levels of H3K9me2 at Chat promoter of females (but, please, see comment 4 in major about the analysis of this variable). However, it is not discussed. Could it be related to the hyperactivity in females compared to males?

Response: We have revised the discussion section to include the following paragraph on weight differences following AIE: “There were also sex specific effects in weight gain during AIE treatment. The effect that AIE-induced suppression weight gain (less than 10%) selectively in males during gavage treatment was not expected, but we did observe a similar phenomenon in a recently published study that also included a larger sample size [50]. This sex differences may be due to differences in ethanol metabolism between the sexes– adult females metabolize ethanol quicker than adult males [51] and may have a briefer intoxication period that could lead to the resumption of typical food consumption in females. However, other work that used a chronic ip injection of ethanol across adolescence found that both male and female rats chronically exposed to ethanol gained less weight gain than saline exposed rats [52]”

Additionally, reanalysis of our H3K9me2 histone methylation data with sex as a factor revealed an overall main effect of exercise irrespective of sex, while H3K9me2 was only increased in males. This is reflected in the discussions section with the following text: “This is the first study that examined epigenetic changes in the cholinergic phenotype following AIE in both male and female rodents. We did find a sex-specific effect: The increased occupancy of H3K9me2 at the Chat promoter was only seen in adult male AIE-treated and was not corrected by exercise. This is contrasted with the sex-independent effect of the upregulation of histone methylation marker H3K9me2 at the more specific region of the Chat promoter CpG island that controls gene silencing, which was corrected by exercise”

9. The conclusion session should be revised. The authors wrote that: “… AIE, its effects on behaviors modulated by the cholinergic system are variable.”. It is difficult to conclude this since there are limitations in the behavioral testes used in this work (lines 451-454; 474-475; 488-495). Besides that, afterwards, the authors address phasic and tonic ACh signaling and attention set shifting behaviors. It was not addressed anywhere else in the manuscript. It is difficult to understand the impact/importance of these types of signaling to the study. If it is important, please provide rationale for it in the introduction and discussion sessions. Response: The conclusion section was rewritten, and we removed the discussion of phasic and tonic ACh signaling.

---

## [Decision Letter · Decision Letter 1]

19 Sep 2024

Voluntary wheel running exercise rescues behaviorally-evoked acetylcholine efflux in the medial prefrontal cortex and epigenetic changes in ChAT genes following adolescent intermittent ethanol exposure

PONE-D-24-25335R1

Dear Dr. Savage,

We’re pleased to inform you that your manuscript has been judged scientifically suitable for publication and will be formally accepted for publication once it meets all outstanding technical requirements.

Kind regards,

Yael Abreu-Villaça, Ph.D.

Academic Editor

PLOS ONE

Additional Editor Comments (optional):

Reviewers' comments:

Reviewer's Responses to Questions

**Comments to the Author**

1. If the authors have adequately addressed your comments raised in a previous round of review and you feel that this manuscript is now acceptable for publication, you may indicate that here to bypass the “Comments to the Author” section, enter your conflict of interest statement in the “Confidential to Editor” section, and submit your "Accept" recommendation.

Reviewer #1: All comments have been addressed

Reviewer #2: All comments have been addressed

2. Is the manuscript technically sound, and do the data support the conclusions?

Reviewer #1: Yes

Reviewer #2: Yes

3. Has the statistical analysis been performed appropriately and rigorously? 

Reviewer #1: (No Response)

Reviewer #2: Yes

4. Have the authors made all data underlying the findings in their manuscript fully available?

Reviewer #1: Yes

Reviewer #2: Yes

5. Is the manuscript presented in an intelligible fashion and written in standard English?

Reviewer #1: Yes

Reviewer #2: Yes

6. Review Comments to the Author

Reviewer #1: All changes made, including separating analyses by sex, lead the manuscript to now be acceptable for publication in PLOS

Reviewer #2: The authors provide a detailed point-by-point response letter, being very responsive to reviewers´ comments. All my concerns were addressed and this version is much improved.

7. PLOS authors have the option to publish the peer review history of their article (what does this mean?). If published, this will include your full peer review and any attached files.

Reviewer #1: No

Reviewer #2: No

---

## [Editor Report · Acceptance letter]

10 Oct 2024

PONE-D-24-25335R1 

PLOS ONE

Dear Dr. Savage, 

I'm pleased to inform you that your manuscript has been deemed suitable for publication in PLOS ONE. Congratulations! Your manuscript is now being handed over to our production team.

Kind regards, 

on behalf of

Dr. Yael Abreu-Villaça 

Academic Editor

PLOS ONE